# The Cinderella Complex: Word embeddings reveal gender stereotypes in movies and books

**Huimin Xu[1], Zhang Zhang[2], Lingfei Wu[3]* , Cheng-Jun Wang[1]* **

**1** School of Journalism and Communication, Nanjing University, Nanjing, China, **2** School of Systems Science, Beijing Normal University, Beijing, China, **3** School of Computing and Information, University of Pittsburgh, Pittsburgh, PA, United States of America

These authors contributed equally to this work.

* wlf850927@gmail.com (LW); wangchj04@126.com (CW)

**Data Availability Statement:** The data underlying the results presented in the study are available from IMDB website (www.imdb.com), IMSDB website (www.imsdb.com), and the Gutenberg Project (www.gutenberg.org). All the code and

## Abstract

Our analysis of thousands of movies and books reveals how these cultural products weave stereotypical gender roles into morality tales and perpetuate gender inequality through storytelling. Using the word embedding techniques, we reveal the constructed emotional dependency of female characters on male characters in stories. We call this narrative structure "Cinderella complex", which assumes that women depend on men in the pursuit of a happy, fulfilling life. Our analysis covers a substantial portion of narratives that shape the modern collective memory, including 7,226 books, 6,087 movie synopses, and 1,109 movie scripts. The "Cinderella complex" is observed to exist widely across periods and contexts, reminding how gender stereotypes are deeply rooted in our society. Our analysis of the words surrounding female and male characters shows that the lives of males are adventure-oriented, whereas the lives of females are romantic-relationship oriented. Finally, we demonstrate the social endorsement of gender stereotypes by showing that gender-stereotypical movies are voted more frequently and rated higher.

## Introduction

Throughout history, stories served not only to entertain but also to instruct. The functions of stories determine their shapes and fates. Among all collectively created stories, including movies, plays, and books, no matter what form they took, those of morals complying with the existing values were more likely to survive. Stories are similar to species in many ways, and the social process for communicating and remembering these mind creatures works as fitness functions to evolve their shapes [1–4]. This process is in favor of stories that reduce the complexity of social lives into memorable and stereotypical descriptions, as these stories are resilient to fast-decaying social attention [5–8]. Dramatic shapes with ups and downs, flat characters, oversimplified causes, all these elements will make a story easier to retell and relate and even become culture memes [3,9]. However, these elements also enhance the spreading of stereotypes broad and far across cultures and periods through storytelling.

processed data are available from https://github.com/xuhuimin2017/storyshape/

**Funding:** This work was supported by the The National Science Foundation of China, grant number 61673070 (http://www.nsfc.gov.cn/) to C-JW. The funder had no role in study design, data collection and analysis, decision to publish, or preparation of the manuscript.

**Competing interests:** The authors have declared that no competing interests exist.

Kurt Vonnegut is among the earliest scholars who proposed to study the shape of stories [10]. Reagan et al. quantified the shape of stories using dictionary-based sentiment analysis. They created a sentiment score dictionary for words, which allowed them to calculate the average scores of sentences that show the ups and downs of stories. [11]. However, their analysis relies on human coders to label the sentiment of words; therefore, it is costly to scale up. We suggest that the emerging word embedding techniques [12,13] provide new tools to automate sentiment labeling and scale up the analysis of stories. Although word embeddings have been used to explore social and cultural dimensions in large-scale corpora [14–16], to our limited knowledge, we firstly apply them to analyze the shape of stories and quantify gender stereotypes.

We firstly construct a vector representing the dimension of happy versus unhappy from pre-trained word vectors using Google News data [17]. The distance from this vector to other word vectors represents the "happiness score" of the corresponding words. The average of "happiness scores" over the timeline of stories quantifies their shape. Moreover, by controlling the window size to analyze only the words surrounding specific names, we can track the "happiness scores" of different characters. Using these techniques, we find that in the movie synopsis of *Cinderella*, the happiness of Ella (Cinderella) depends on Kit (Prince) but not vice versa. This finding supports the "Cinderella complex" [18], a narrative structure enhancing the stereotypical incompetence of women. Applying our analysis to 6,087 movie synopses, 1,109 movie scripts, and 7,226 books, we observe the vast existence of this narrative structure. Our review of the words surrounding characters unpack their stereotypical life packages; the lives of males are adventure-oriented, whereas the lives of females are romantic-relationship oriented. Finally, we reveal the social endorsement of gender stereotypes by identifying the association between the strength of gender stereotypes in movie synopses and the IMDB ratings to the analyzed movies.

## Gender stereotypes: Women's lacking of competence and agency

According to the research of social psychology, gender stereotypes are inaccurate, biased, or stereotypical generalizations about different gender roles [19]. It is associated with limited cognitive resources and people's tendency to overstate the differences between groups yet underestimate the variance within groups [19]. The gender roles originate from labor division historically [20] and diffuse into many other social dimensions [21], including education, occupation, and income [21,22]. One significant consequence of gender stereotypes is the reinforcement of gender inequality through parenting styles and conventions in school and workplace [23,24].

As one of the most pervasive stereotypes, gender stereotypes reflect the general expectations about the social roles of males and females. For example, females are communal, kind, family-oriented, warm, and sociable, whereas men should be agentic, skilled, work-oriented, competent, and assertive [19,25]. Many scholars argued that agency versus communion is the primary dimension to study [26,27], while some others emphasized competence instead of agency [28]. Cuddy et al. found that agency and competence tend to be correlated [28].

The rich literature on gender stereotypes points out the assumptions to explore in identifying and quantifying stereotypical narrative structures, including 1) *The emotional dependency of females on males*. Men and women have different social imagines. Men are agentic, and women are communal [19,29,30]; men are active, whereas women are passive; men give, and women take in relationships [29–31]. These biased images of men and women lead to biased expectations in their relationships. Those who consider women less competent would tend to believe that they are fragile and sensitive and need to be protected by men [19,32]. Following

this literature, we propose to test the emotional dependency of females on males [33]; 2) *Men act and women appear*. The English novelist John Berger [34] used this quote to describe the male-female dichotomy. Considering the stereotypical role and traits of men, one would imagine men are more likely than women to be described using verbs; 3) *Social endorsement of gender stereotypes*. The social and cultural roots of gender stereotypes form social force against stereotype disconfirmation from people, action, or ideas [19]. In this sense, the stories that approve gender stereotypes will gain social approval themselves, whereas the stories against stereotypes will be ignored and disapproved. Our study will test this assumption by connecting the frame of stories with their social acceptance.

While existing literature primarily focuses on stereotype reinforcement through people and actions [19], Colette Dowling's analysis reveals stereotypes in ideas and narratives. The term Cinderella complex first appeared in Colette Dowling's book *The Cinderella Complex*: *Women's Hidden Fear of Independence*. It describes women's fear of independence and an unconscious desire to be taken care of by others [18]. For example, in the story *Cinderella*, Ella is kind-hearted, beautiful, attractive, and independent, yet cannot decide her own life and has to rely on the support from others (e.g., the fairy godmother), especially the male characters (e.g., Kit the Prince). Dowlings argued that the story of *Cinderella* amplified the psychological and physical differences between women and men and implied that women depend on men in the pursuit of a happy, fulfilling life. In this paper, we present evidence supporting Dowling's arguments on the emotional dependency of females on males in our analysis of the movie synopsis of *Cinderella*. We also examine the other two assumptions on stereotypical narrative structures using machine-enhanced analysis of large corpora.

## Results

### 1. The Cinderella complex: Men are women's ways to happiness

We select the text of *Cinderella* from the movie synopsis data, which contains 97 sentences. Within each sentence, we calculate the distance from the pre-trained vectors of words [17] to the constructed happiness vector to derive the happiness scores of words. The emotional status of characters is built up by a sequence of events; therefore, we sum the scores across sentences alongside the story timeline to measure the cumulative average rather than the marginal variance of emotion. For example, the lowest point of the emotional curve of Cinderella (at around 30 percentile of the story timeline in Fig 1B) is the consequence of the death of parents and maltreatment from the stepmother. Therefore, the emotional status of Cinderella at this stage is better measured by the cumulative score rather than the score of any single sentence.

The story begins with a happy life of Ella with both parents. The death of her mother is associated with the first drop of the emotion curve (around five percentiles on the x-axis in Fig 1B). The reorganization of the family and the death of her father on a business trip make her life bumpy. Under the maltreatment of the stepmother, Ella's life goes all the way down to the bottom (30 percentiles), but this is also when the twist happens—Ella meets Kit in the forest. After the upsetting separation, joining the royal ball party and dancing with Kit with the help from the fairy godmother makes the emotion curve peak (65 percentiles). Leaving the party and losing the crystal shoes pull the curve down again, but the reunion with Kit pushes the curve back (90 percentiles).

In contrast, the emotion curve of Kit is less bumpy, especially at the scenarios he interacts with Ella, i.e., the sentences that contain both names. In other words, the happiness of Ella is driven by Kit, whereas the happiness of Kit is relatively less elastic to their interaction. These findings present evidence for Dowling's analysis on "Cinderella complex", the dependency of females on males in the pursuit of a happy life [18].

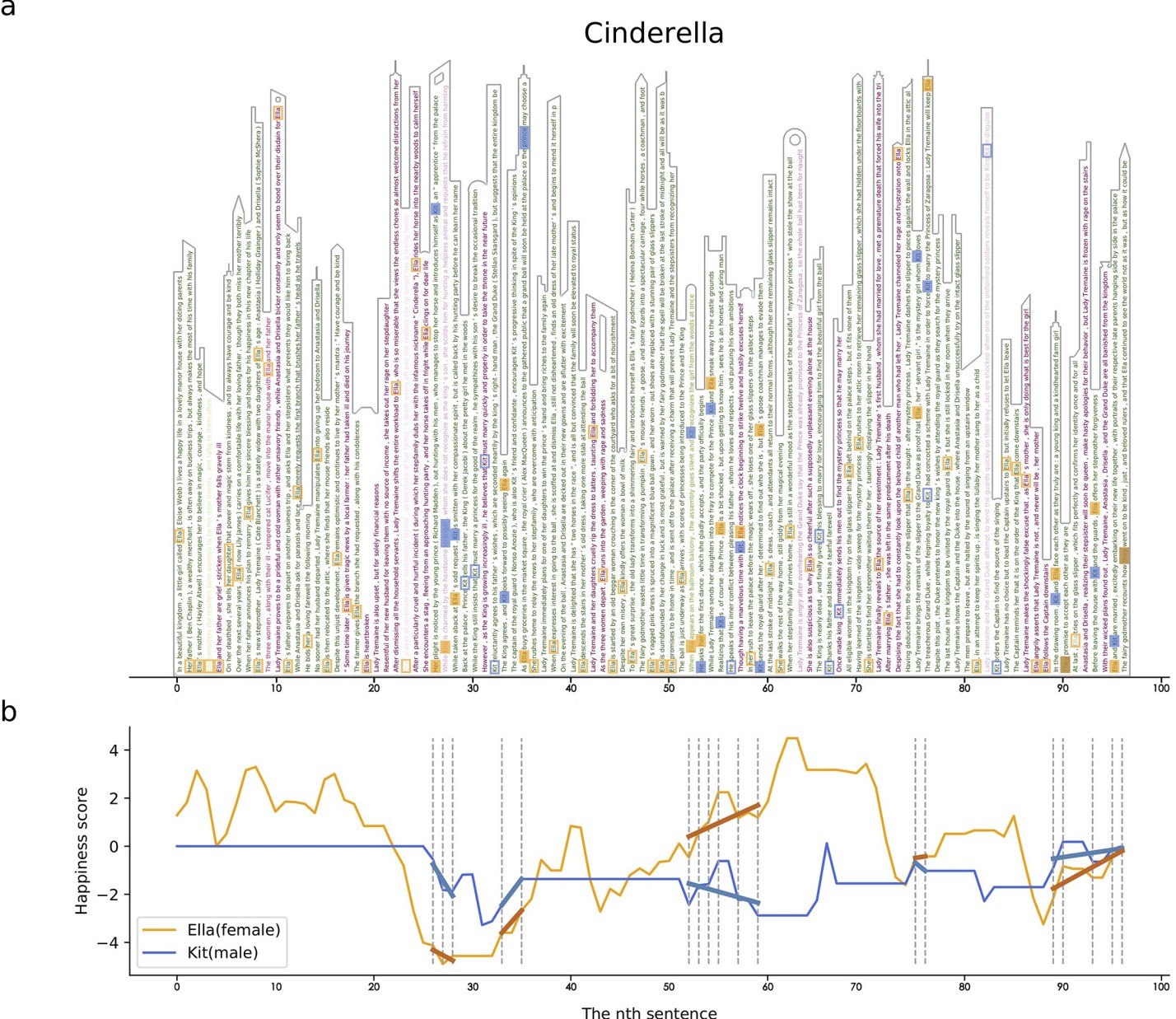

**Fig 1. The Cinderella complex. a**, Visualizing the sentiment landscape of the movie synopsis of *Cinderella* as a skyline (a black outline is added to enhance the "skyline"metaphor visually). We show sentences in a vertical schema, colored by their "happiness score"—green for happy and red for unhappy and the transparency represents the scale of the scores. Filled squares (orange for Ella and blue for Kit) indicate the co-occurrence of Ella (Cinderella) and Kit (the Prince) in the same sentence. Hollow squares indicate where only one character appears: **b**, The happiness curves of Ella (orange) and Kit (blue). The grey dotted lines marks the sentences in which they co-occur, corresponding to the filled squares in Panel **a**. We fit the increase or decrease in happiness scores across successive co-occurrences with OLS regression (see Methods for more information). Thick lines show the estimates of regression.

The naturally following question is, how general is this pattern? Is it as Dowling predicted—existing widely across periods and contexts? We select ten movies across genres, lengths, periods, and the gender of the leading characters (defined by the most frequently observed name in the text). We find the asymmetric emotional dependency of females on males across these ten movies—Cinderella is not the only character who has complicated feelings on males (Fig 2, books see Fig 3).

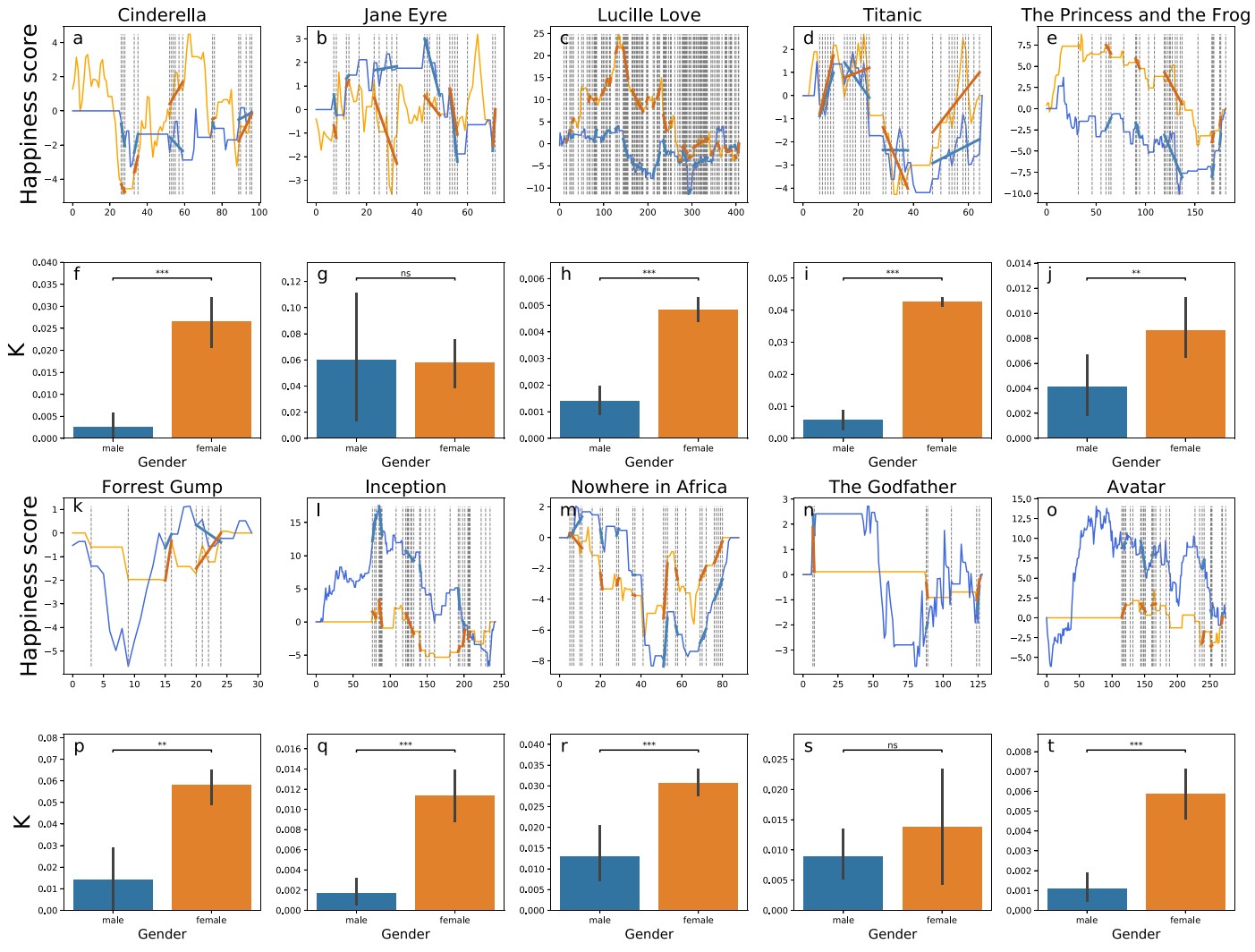

**Fig 2. The Cinderella complex across ten movie synopses. a-e**, The happiness curves for five movie synopses in which the leading character is female. We define happiness curves in the same way as in Fig 1B. **f-j**, The increase in happiness conditional on the co-occurrence with the other gender, measured in the average of positive regression coefficients k, are shown as bars (blue for males and orange for females). The lines on the top of the bars show one standard deviation. Asterisks indict P values. * P ≤ 0.05, ** P ≤ 0.01, *** P ≤ 0.001, and ns non-significant. **k-o**, The happiness curves for five movie synopses in which the leading character is male. Panels designed in the same schema as **a-e**. **p-t**, The increase in happiness conditional on the co-occurrence with the other gender. Panels designed in the same schema as **f-j**.

We further examine this pattern across 7,226 books, 6,087 movie synopsis, and 1,109 movie scripts. We find that despite the association between females and emotional words (see S1 Fig for this association), when male and female characters are present in the same context (which is a set of sequential sentences, see the definition in Method), the happiness score of female characters is significantly higher than that in the contexts without the presence of male characters (S2 Fig). Meanwhile, within the context of character occurrence, the increase in the happiness score is higher for female characters than for male characters (Fig 4). These findings reveal a constructed, asymmetrical emotional dependency of females on males, robust against the publication time (S7 Fig) and genres (S8 Fig) of stories and the direction of emotion (S3 Fig).

## 2. Men act and women appear: A life unpacked

To unfold the constructed contexts justifying the emotional dependency of female characters on male characters, we analyze the words surrounding the names of characters. Within each of the 6,087 movie synopses, we select five words before and five words after the names of the

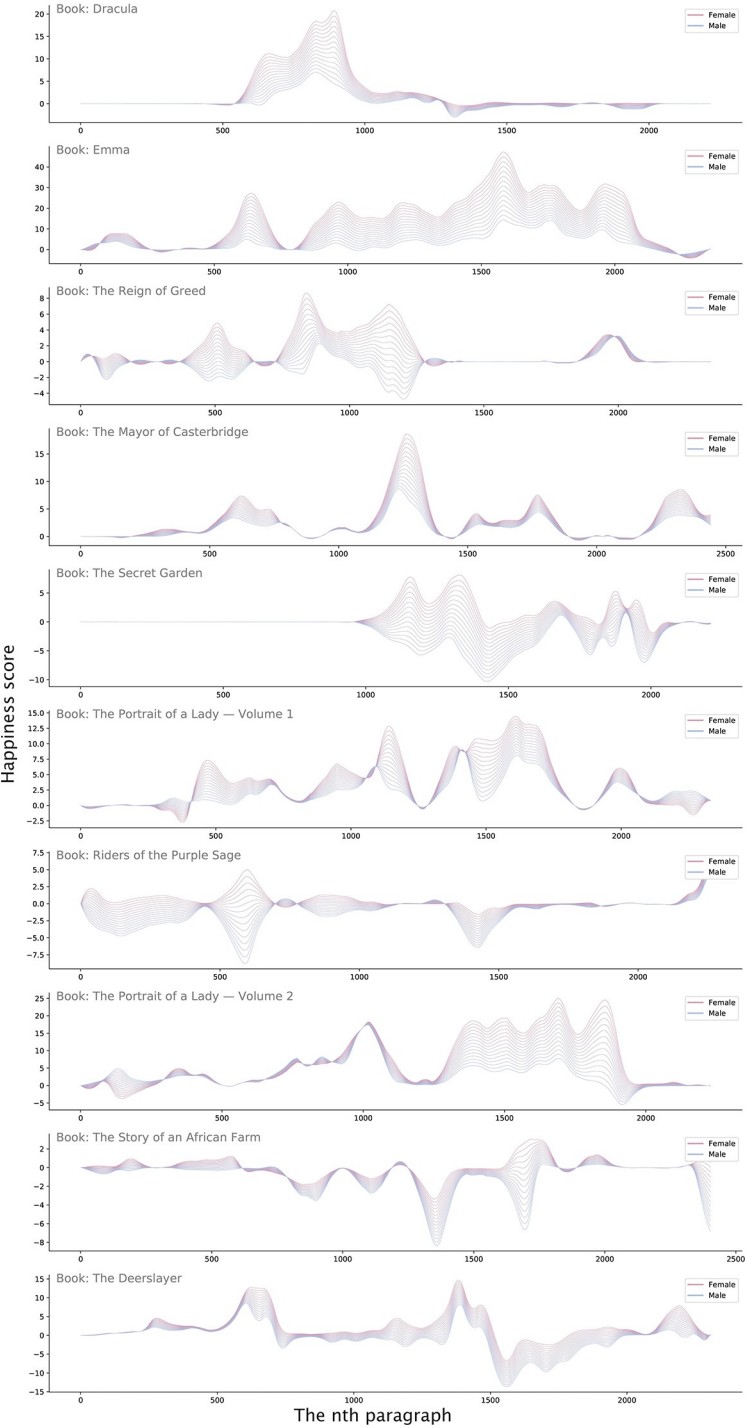

**Fig 3. The Cinderella complex across ten books.** The code is available on https://github.com/wenoptics/viz-of-gender-stereotypes-from-texts/.

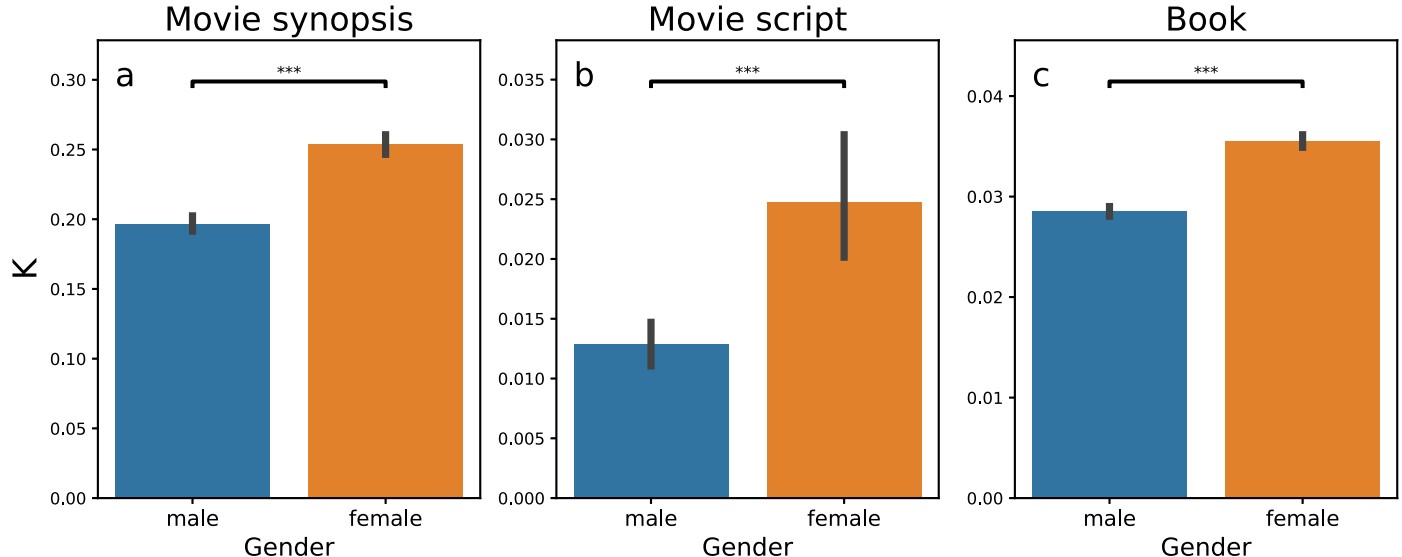

**Fig 4. The Increase in happiness, conditional on the co-occurrence with the other gender, is higher for female than for male characters.** We analyze three datasets, including 6,087 movie synopses (**a**), 1,109 movie scripts (**b**), and 7,226 books (**c**). The increase in happiness conditional on the co-occurrence with the other gender, measured in the average of positive regression coefficients k, are shown as bars (blue for males and orange for females). The result of decrease in happiness is shown in S3 Fig. The lines on the top of the bars show one standard deviation. Asterisks indict P values. * $P \leq 0.05$, **$P \leq 0.01$, *** $P \leq 0.001$, and ns non-significant. The result is significant across the three datasets.

leading characters across all the sentences. We iterate over the pairwise combinations of words within 10-word samples across all movie synopses to construct word co-occurrence networks, one for females and the other for males (Figs 5 and 6). We then identify communities from

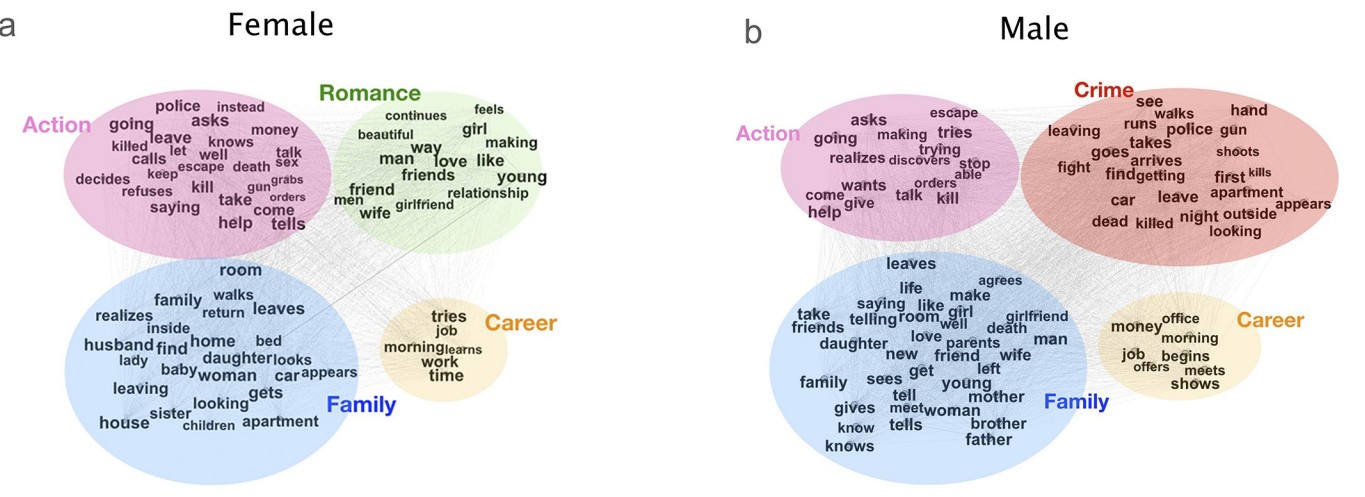

**Fig 5. Word co-occurrence networks describing the life packages of female vs. male as leading characters.** For each of the 6,087 movie synopses under study, we select ten words surrounding the names of the leading characters (five words before and five words after) across all the sentences containing the names. We iterate over the pairwise combinations of words within each 10-word sample across all movie synopses to construct word co-occurrence networks, one for males and the other for females. The female network (a) has 39,284 nodes (words), and 921,208 links (pairwise combinations of words within samples) and the male network (b) has 46,909 nodes and 1,319,208 links. We detect communities from the networks using the modularity algorithm [35]. Top four communities emerge from both networks, including action, family, career, and romance in the female network and action, family, career, and crime in the male network. Only nodes of 1,500 or more links are labeled.

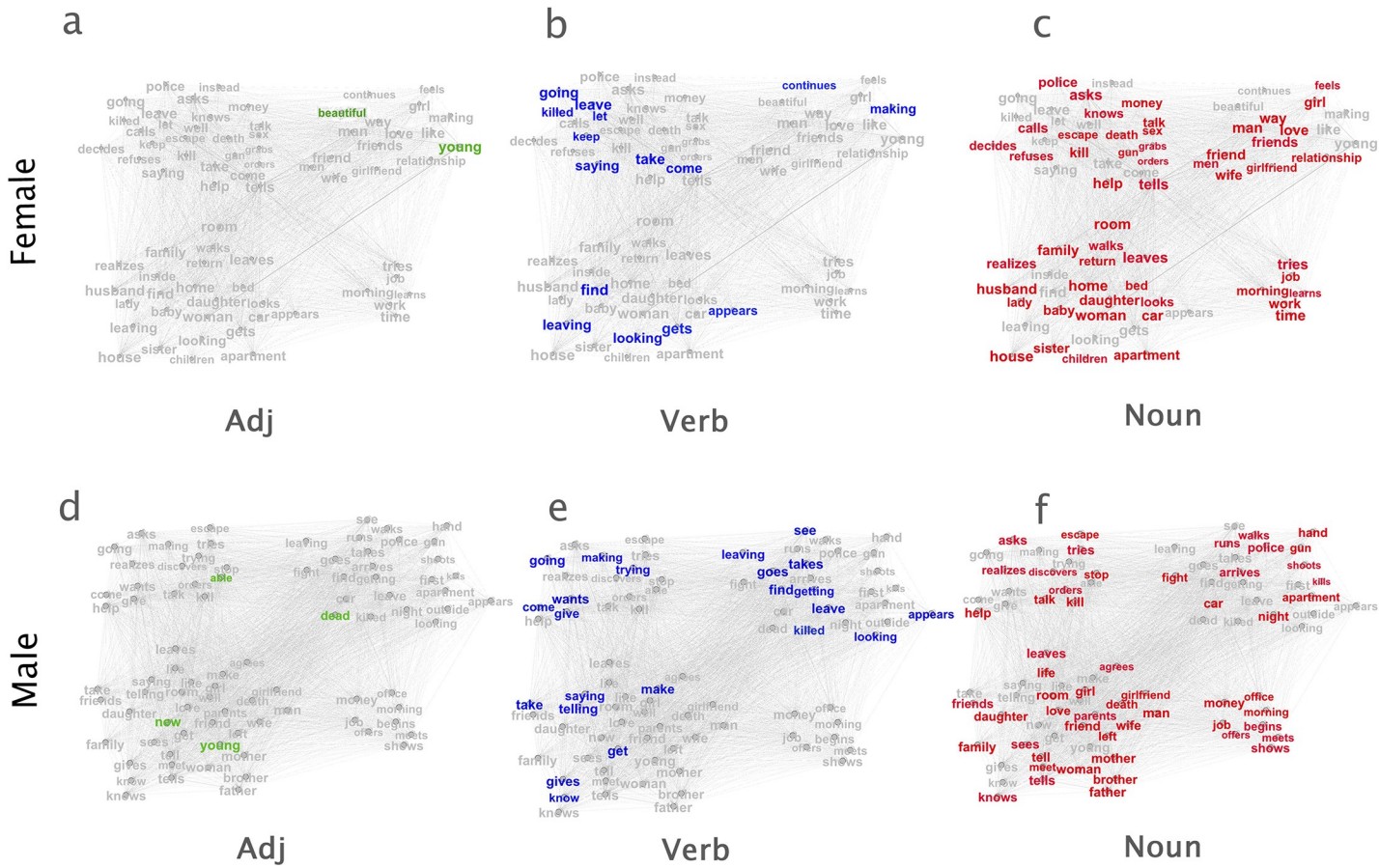

**Fig 6. The distribution of adjectives, verbs, and nouns in female and male word co-occurrence networks. a-c.** The distribution of adjectives (**a**, green labels), verbs (**b**, blue labels), and nouns (**c**, red labels) in the female word co-occurrence network as introduced in Fig 5A. **d-f.** The distribution of adjectives (**a**, green labels), verbs (**b**, blue labels), and nouns (**c**, red labels) in the male word co-occurrence network as introduced in Fig 5B. Word categories are detected using the Penn Treebank tagset [36]. Only nodes of 1,500 or more links are labeled.

these two networks using the Q-modularity algorithms [35]. Four communities emerge from both networks, including action, family, career, and romance in the female network and action, family, career, and crime in the male network (Fig 5). This community structure reveals romantic-relationships define females characters, while adventures and excitements build males characters.

We further cut both networks into three slices by word categories, including adjectives, verbs, and nouns. The differences in the distribution of words portray stereotypical gender images in detail. For example, both females and males may be described as "young", but females are more likely to be "beautiful", and males are more likely to be "able". Further, we observed that male characters are more likely than female characters to be described using verbs across three datasets (Fig 7). This observation reminds the quote, "Men act, and women appear" from the English novelist John Berger [34]. He used this quote to summarize the stereotypical ideas that men are defined by their actions, whereas women are defined by their appearances. Aside from analyzing all the words describing the male and female characters (Figs 5–7), we also select and analyze the words within the context of the presence of both genders. The observed difference between groups in Figs 5–7 remains significant (see S4–S6 Figs for more information). In sum, our analysis reveals the stereotypical male-female dichotomy

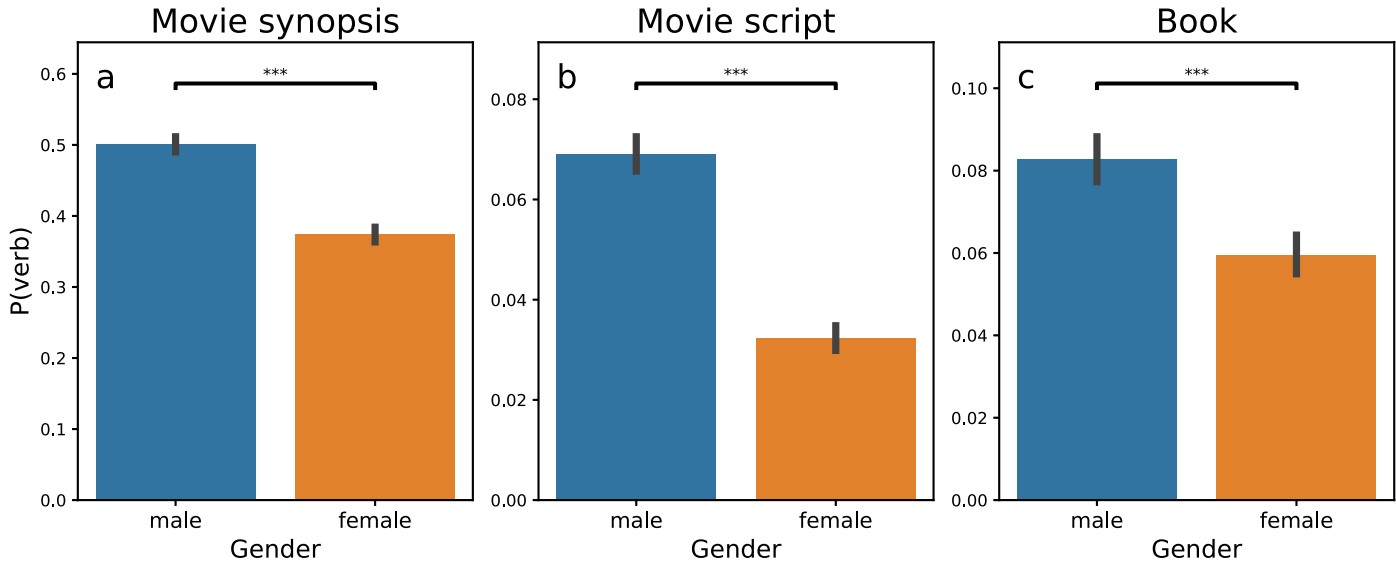

**Fig 7. Men are more likely than women to be described using verbs.** We analyze three datasets, including 6,087 movie synopses (a), 1,109 movie scripts (b), and 7,226 books (c). For each movie synopsis, movie script, or book under study, we select ten words surrounding the names of the leading characters (five words before and five words after) across all the sentences containing the names. We detect word categories using the Penn Treebank tagset [36] and calculate the probability of observing verbs, P(verb), across all 10-word samples for females or males within each dataset. The bars show the values of P(verb), blue for males and orange for females. The lines on the top of the bars show one standard deviation. Asterisks indict P values. * $P \leq 0.05$, **$P \leq 0.01$, *** $P \leq 0.001$, and ns non-significant. The result is significant across the three datasets.

that females are communal, kind, family-oriented, warm, and sociable, whereas men should be agentic, skilled, work-oriented, competent, and assertive [19,25].

## 3. The Cinderella complex: People's choice

We assume that there is a hidden culture market that suppresses stereotype disconfirmation from people, action, or ideas [3,19,37] and approves stereotype reproduction from all these directions. We explore how the strength of Cinderella complex in narratives, measured by the level of emotional dependency of females on males, is associated with the acceptance of narratives. We use the averaged ratings to movies, which vary from 0 to 10, as a proxy of movie reputation. The number of votes, defined as the number of audiences who rate the movie, measures movie popularity. These two variables characterize the cultural market shaped by social choices.

We observed that the increase in female happiness conditioning on co-occurring with male characters has a positive impact on both the ratings and popularity of movies (Table 1). In contrast, the increase in male happiness has a negative influence on movie acceptance. In other words, narratives presenting the emotional dependency and vulnerability of females are perceived as "good stories", but movies highlighting the emotional vulnerability of males are not as much welcomed. These results are robust to story intensity (the number of sentences with the co-occurrence of female and male characters) and the gender of the leading character (Table A and Table B in S1 File). For independent variables, the leading gender means the gender of the main character who dominantly shows up in the movie script, where the male is 1 and female is 0. N of co-occurrence implies the number of sentences that the characters of both genders appear in the same sentence. We found that story intensity (the number of sentences with the co-occurrence of female and male characters) and story with male leading character also have a positive impact on both the ratings and popularity of movies.

**Table 1. OLS regressions predicting the number of votes and rating in the movie synopses dataset.**

|  | Rating | N of Votes |
|---|---|---|
| **Constant** | 6.12*** | 8.11*** |
| **The gender of the leading character (male = 1, female = 0)** | 0.18*** | 0.38*** |
| **N of sentences with the co-occurrence of female and male characters** | 0.01* | 0.02*** |
| **Increase in happiness for female characters** | **0.06**** | **0.09*** |
| **Increase in happiness for male characters** | **-0.08**** | **-0.29**** |
| **R-squared** | 0.016 | 0.048 |
| **F-statistic** | 13.03 | 40.10 |
| **N of cases** | 6,087 | 6,087 |

Note. Asterisks indict P values.

* $P \leq 0.05$

** $P \leq 0.01$, and

*** $P \leq 0.001$.

## Conclusions and discussions

After three waves of feminism [38], words like brave and independent are more likely to associate with female roles [39]. Females' increasing entry into professional occupations enhances their perceived competence, and the improvement of their education level also helps break the gender stereotypes [25]. In a recent study, Gard et al. analyzed gender stereotypes in the past century using word embeddings and found that gender bias was decreasing, especially after the second-wave feminism in the 1960s [15]. The meta-analysis based on 16 U.S. public opinion polls (1946–2018) showed that social expectation on the competence and intelligence of females increased over time, but the expectation on the agency of females remained low [26]. This observation is consistent with our analysis of the passive and agency-lacking female characters.

Our study, while primarily focuses on designing and testing existing assumptions on gender stereotypes, also aims to contribute to the theories on gender stereotypes in several dimensions. 1) Interacting vs. separated gender roles. The analysis of the relationships between genders is critical to reveal stereotypical expectations, as gender roles emerge from the interactions with the other gender. 2) Visible vs. hidden stereotypes. Some gender inequalities and stereotypes are more noticeable than others, such as inequalities in voting rights, working salaries, and educational opportunities. These apparent inequalities may distract social attention and make hidden stereotypes in paradigms, language, and communication even less noticeable [30]. 3) Social reproduction of stereotypes. There are both causes and consequences of stereotypical narratives. Stereotypes reduce the complexity of stories and make them more relatable and memorable; however, the flat characters may project into reality. Gender stereotypes, constructed and weaved into the moral tales from movies and books, may maintain gender inequality though these morality norms and reproduce gender inequality as a social fact [23]. For example, when children are exposed to stereotyped narratives, they may fill themselves into stereotypical roles [40]. A study on the impact of Disney movies shows that children who associate beauty to popularity for movie characters tend to apply the same principle in real lives [41].

The limitations of the current study are noted and should be aware of in future research designs. The natural language processing models used to identify the leading characters their gender (www.nltk.org/book/ch02.html) may miss the uncommon names of characters or

misidentify characters genders. Also, there is an unexplained variance between machine-labeled versus human-labeled happiness scores for words (Pearson correlation coefficient equals 0.53 with a P-value < 0.001). In general, sentiment scores for words have limitations in analyzing narratives as a fixed score, since they can not capture the variance of sentiments of the same word across contexts.

## Methods

### Data collection

We collect three datasets for this present research, including movie synopses, movie scripts, and books (Fig 8). We collect the movie synopsis data from the IMDB website (www.imdb.com). We select the movies with user ratings, plot synopsis, release year, and genre. And we get 16,255 movies for further data filtering. We choose 6,087 movie synopses with more than five sentences and both female and male characters in the analysis. Second, we also collect the movie script data from the IMSDB website (www.imsdb.com), which is the largest database of online movie scripts. There are 1,109 movie scripts after filtering out those in which only one

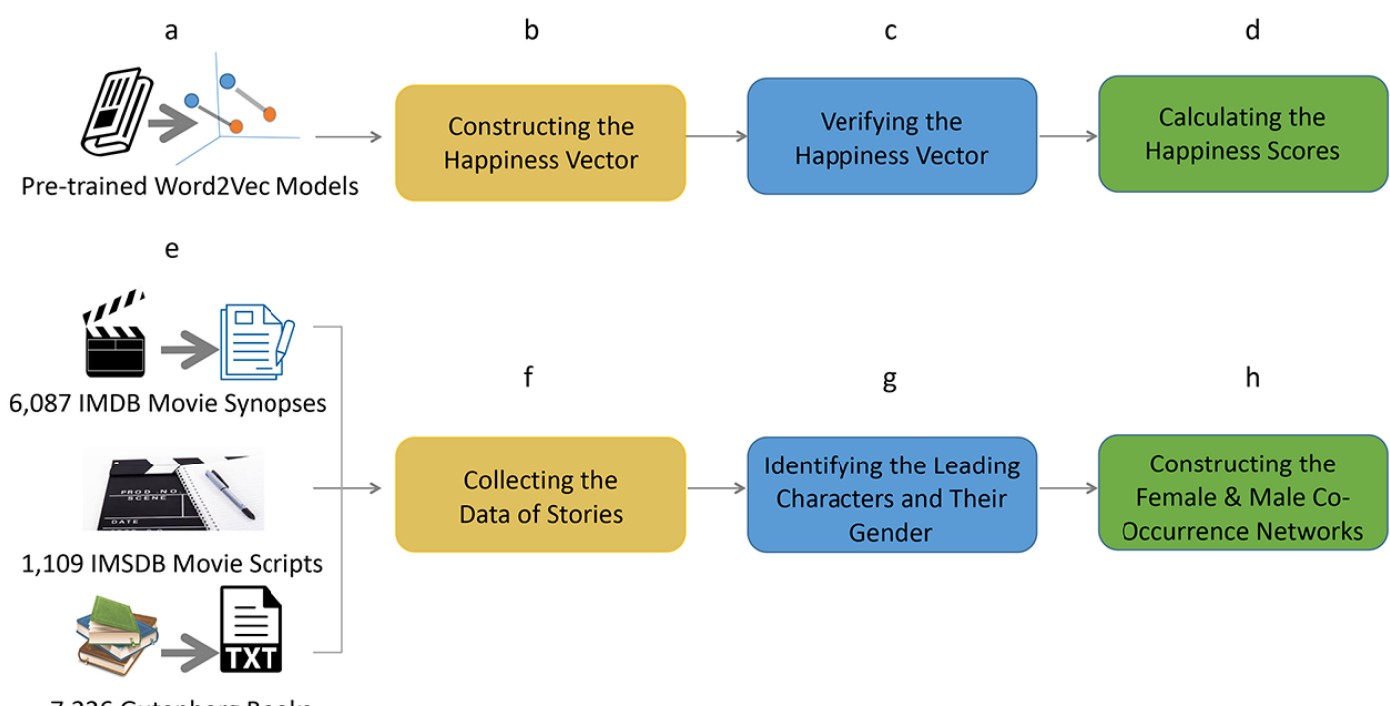

**Fig 8. Data collection and cleaning. a,** Pre-trained 300-dimension word embeddings using Google News [17]. **b,** We select two sets of words, one for positive sentiment and the other for negative sentiment. We then subtract the average vector of the negative words from the average vector of the positive words to obtain the "happiness vector" [14]. **c,** The constructed happiness vector is verified using a human-labeled dataset. We select 10,000 words from the Hedonometer project (http://hedonometer.org/words.html), each of which was assigned a happiness score ranging from one to nine by Amazon's Mechanical Turk workers [42]. The distance from the Google News vectors of these 10,000 words to our "happiness vector" is positively correlated with their manually assigned happiness score. The Pearson correlation coefficient equals 0.53 (P-value <0.001). **d,** We calculate the happiness score of each word in the analyzed text by measuring the cosine distance from their Google News vectors to the constructed happiness vector. **e-f,** Three datasets in this study, including 6,087 movie synopses, 1,109 movie scripts, and 7,226 books. **g,** For each dataset, the leading characters and their gender are identified to track their emotional fluctuation. **h,** Word co-occurrence networks are constructed to describe the life packages of female vs. male as leading characters. These networks contain words surrounding the names of the leading characters (with a window size of ten words) as nodes and their pairwise combinations as links.

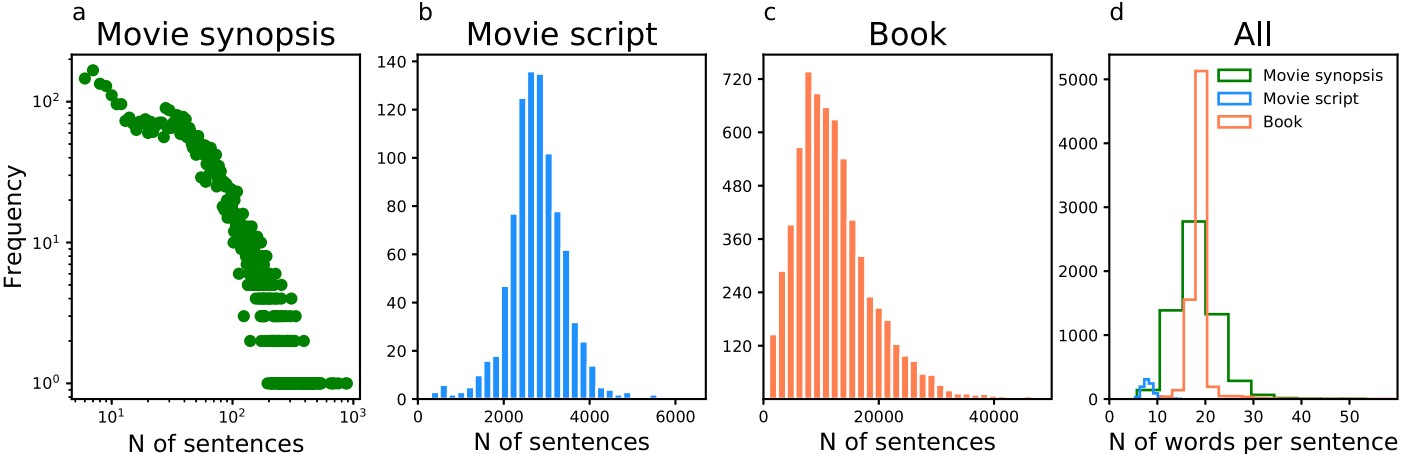

**Fig 9. Story length in sentence.** The distribution of the number of sentences of 6,087 movie synopses (**a**), 1,109 movie scripts (**b**), and 7,226 books (**c**). **d**, The number of words per sentence across three datasets.

gender of characters are identified. The metadata of the movie scripts, such as the release year and genre, is also collected. Third, in addition to the two movie datasets, we also collect the data of more than 40 thousand English books from the Gutenberg Project ([www.gutenberg.org](http://www.gutenberg.org)), including the text of story, publication time, and genre. In the data filtering of books, only 7,226 books belonging to the genre "language and literature" and containing both female and male characters are selected. All the code and data are available from [https://github.com/xuhuimin2017/storyshape/](https://github.com/xuhuimin2017/storyshape/).

Fig 9 compares the length of stories in sentence across three datasets. Since the users of IMDB website create the movie synopses, the variance in story length is much more significant than that in movie scripts and books, as the scripts and books are typically from a smaller group of authors. Because the length of dialogues is usually short, the average number of words per sentence for the movie script data is much smaller than the other two datasets. Given the different number of sentences in three datasets, we segment movie synopses by sentences, while segment movie scripts and books by paragraphs. Since the sentence is the primary unit of narrative, this method of story segmentation helps us understand the variance of sentiment in stories.

## Constructing the happiness vector and calculating happiness scores

Frame analysis proposed by Goffman is widely used to analyze the structure of narrative and reveal its bias [43]. A frame is a scheme of interpretation to organize the details of events and human behaviors. It could be a set of stereotypes working as cognitive "filters" for making complex social realities easy to interpret. Framing has consistently been shown to be an influential source of social bias in decision-making [44]. Framing involves four key steps: "define problems, diagnose causes, make moral judgments, and suggest remedies" [45]. To frame the identities of social roles is a typical approach for building stereotypes around underprivileged groups so as to justify unfair social systems [46]. For example, Iyengar argues that episodic television frames tend to blame the poor themselves for poverty, compared with the thematic television news frames [47]. However, despite the importance of frame analysis in revealing the formation of social bias, its limitation is also apparent. Frame analysis originates from and is strongly influenced by rhetorical analysis, which tends to amplify all rhetorical details of narrative and may lose the focus of the massive structure. Also, frame analysis involves content

analysis conducted by human coders who are trained to label the content using codebooks manually. It is costly in time and human research workforce and hard to scale up and validate.

The advances in natural language processing (NLP) techniques and availability of large scale text data unleash tremendous opportunities to automate frame analysis of stories and study gender stereotype. According to the study of Caliskan et al., the word embeddings derived from text corpora can also be used to reveal the stereotypes of stories [14]. Caliskan et al. show that the fraction of female workers within each occupation is strongly correlated with the Cosine distance from the vector representing female to the vector representing occupation [14]. Garg et al. use word embeddings trained on the text data of 100 years to capture the evolution of gender bias over time. They find that from 1910 to 1990, the measured gender bias was decreasing [15]. Using a similar method, Kozlowski et al. show that in addition to occupations, gender bias also exists widely in sports, food, music, vehicles, clothes, and names [16].

We propose to use word embedding techniques for the analysis of gender stereotypes. Word embeddings provide a better solution to analyze the sentiments of text and to deal with the high dimensional semantic relationships between words [12]. Instead of relying on human-labeled sentiment scores, the word embedding method constructs the emotion vector and calculate the emotion score for every word in the document automatically. Therefore, it is more fine-grained compared with the emotional dictionary method. The accuracy of sentiment analysis can be significantly improved using the word embedding method [12,13]. There are several publicly accessible datasets, including 300-dimension Google News vectors [13,17], 300-dimension Wikipedia and Gigaword vectors [48,49], and 200-dimension Twitter vectors [48,49].

To compare these word embeddings and choose the best word embeddings for our analysis, first, we construct a vector representing "happiness" by retrieving the pre-trained embedding vectors of two sets of words, including success, succeed, luck, fortune, happy, glad, joy, smile for positive and failure, fail, unfortunate, unhappy, sad, sorrow, tear for negative sentiment. By subtracting the average vector of the positive words from the average of the negative words, we created the "happiness" vector using these pre-trained vectors. Second, we use the happiness scores of 10,000 sentiment words provided by the Hedonometer project (http://hedonometer.org/words.html). By merging the 5,000 most frequently used words from Google Books, New York Times articles, Music Lyrics, and Tweets, Dodds et al. got these 10,000 words. Each of these words was assigned a happiness score ranging from one to nine by Amazon's Mechanical Turk workers [42]. We get the word vectors for the 10,000 words using these pre-trained vectors and calculate the Cosine distances between each of these 10,000 words vectors and the happiness vector. We compute the Pearson correlation coefficients between the Cosine distances of these 10,000 words and their happiness scores. It turns out the Pearson coefficient calculated with Google News embeddings is the largest (0.53***), compared with the person coefficients computed with Wiki & Giga embedding (0.40***) and Twitter embedding (0.47***). Therefore, in this study, we employ the pre-trained word vectors trained on Google News dataset for our analysis.

To obtain the emotion curves of characters, we firstly get the happiness score of each word by calculating the distance from their Google News vectors to the constructed happiness vector. Then, we can obtain the happiness scores averaged for each sentence or paragraph and normalize the happiness scores within a character with Z Score method. For two characters in the same context, we assume that they share the same raw scores of happiness. To better measure the happiness score for different characters over time, the happiness score of the sentence or paragraph without the name of either female or male character is 0. In this way, we can get the happiness curve of different characters for the whole story. We finally accumulate the

happiness curve across sentences or paragraphs that contain the names of either female or male character to get the overall emotion trend. Take the story of Cinderella as an example (Fig 1A), when Ella and Kit get married at the end of the story, they share the same raw happiness score estimated from the sentence. But for Ella, this score results in a more positive pull in the displayed curve than that of Kit, because Ella was upset of her stepmother before the marriage, which makes the raw happiness score from marriage translated into the higher z-score.

## Identifying the leading characters and their gender

To investigate how the other gender influences the leading characters, we need to identify character names and their gender. The IMDB dataset provides the information of the main cast that includes the gender information (in the form of "actor" or "actress"), cast names, and character names. The movie script dataset contains the dialogues between characters (put the character name before the dialogue), which can also help us to identify the person names in stories. We then employ a pre-trained gender classifier (github.com/clintval/gender-predictor) to predict the gender of the character names. In the book dataset, we use the names corpus for males and females from the NLTK package (www.nltk.org/book/ch02.html) to identify name and gender together. Also, we use the *neuralcoref* package in Python to annotate and resolve the coreference clusters (huggingface.co/coref). To identify the leading character, we count the frequency of person names appeared in stories. For example, if the most common name is female, then it is a female-dominated story and vice versa. Finally, we measure the co-occurrence of male and female character by finding whether they appear together in the same sentence for movie synopses or in the same paragraph for movie scripts and books.

## Measuring the increase and decrease in happiness during the co-occurrence

We measure the increase or decrease in happiness scores with OLS regression. First, we normalize the emotion curve to the range from 0 to 1 to compare the slopes across different characters in different stories. Then, we fit regression models to the happiness curves across successive co-occurrences between male and female characters. In this way, we can get the slopes with regression coefficients to measure the increase and decrease in happiness scores. To be specific, the increase in happiness is measured by the average of positive OLS regression coefficients and weighted by the sample size (i.e., number of sentences or paragraphs in the regression). And the decrease in happiness is measured by the average of negative OLS regression coefficients and weighted by the sample size. Also, we merge the nearby sentences or paragraphs of co-occurrences. Therefore, it is necessary to consider the different length of the gap between sentences or paragraphs of co-occurrence. After merging the sentences or paragraphs of co-occurrence in chronological order using different gap length (ranging from 1 to 10), we find that the results are robust.

## Supporting information

**S1 Fig. The overall happiness level, is higher for female than for male characters.** We analyze three datasets, including 6,087 movie synopses (**a**), 1,109 movie scripts (**b**), and 7,226 books (**c**). The happiness score averaged over the whole course of stories are shown as bars (orange for females and blue for males). The lines on the top of the bars show one standard deviation. Asterisks indict P values. * P ≤ 0.05, **P ≤ 0.01, *** P ≤ 0.001, and ns non-significant. The result is significant across the three datasets.
(EPS)

**S2 Fig. Females are happier when they encounter (co-occur in the same sentence) males.**
We analyze three datasets, including 6,087 movie synopses (**a**), 1,109 movie scripts (**b**), and 7,226 books (**c**). Bars show the happiness scores, orange for co-occurring with males and blue otherwise. The lines on the top of the bars show one standard deviation. Asterisks indict P values. * $P \leq 0.05$, **$P \leq 0.01$, *** $P \leq 0.001$, and ns non-significant. The result is significant across the three datasets.
(EPS)

**S3 Fig. The decrease in happiness, conditional on the co-occurrence with the other gender, is higher for female than for male characters.** We analyze three datasets, including 6,087 movie synopses (**a**), 1,109 movie scripts (**b**), and 7,226 books (**c**). The decrease in happiness conditional on the co-occurrence with the other gender, measured in the average of negative regression coefficients $k$, are shown as bars (blue for males and orange for females). The lines on the bottom of the bars show one standard deviation. Asterisks indict P values. * $P \leq 0.05$, **$P \leq 0.01$, *** $P \leq 0.001$, and ns non-significant. The result is significant across the three datasets.
(EPS)

**S4 Fig. Word co-occurrence networks describing female vs. male when they meet the other gender.** For each of the 6,087 movie synopses under study, we select ten words surrounding the names of the leading characters (five words before and five words after) across all the sentences containing both names of the female and male leading characters. We iterate over the pairwise combinations of words within each 10-word sample across all movie synopses to construct word co-occurrence networks, one for males and the other for females. The female network (**a**) has 9,379 nodes (words), and 73695 links (pairwise combinations of words within samples) and the male network (**b**) has 13,776 nodes and 225,473 links. We detect communities from the networks using the modularity algorithm [35]. Three communities emerge from the female network, including action, family, and romance. And five communities are identified from the male network, including action, family, romance, crime, and career. Only nodes of 500 or more links are labeled.
(TIFF)

**S5 Fig. The distribution of adjectives, verbs, and nouns in word co-occurrence network. a-c.** The distribution of adjectives (**a**, green labels), verbs (**b**, blue labels), and nouns (**c**, red labels) in the female word co-occurrence network as introduced in S4A Fig. **d-f.** The distribution of adjectives (**a**, green labels), verbs (**b**, blue labels), and nouns (**c**, red labels) in the male word co-occurrence network as introduced in S4B Fig. Word categories are detected using the Penn Treebank tagset [36].
(TIFF)

**S6 Fig. Men are more likely than women to be described using verbs on the co-occurrence with the other gender.** We analyze three datasets, including 6,087 movie synopses (a), 1,109 movie scripts (b), and 7,226 books (c). For each movie synopsis, movie script, or book under study, we select ten words surrounding the names of the leading characters (five words before and five words after) across all the sentences containing both names of the female and male leading characters. We detect word categories using the Penn Treebank tagset [36] and calculate the probability of observing verbs, P(verb), across all 10-word samples for females or males within each dataset. Bars show the values of P(verb), blue for males and orange for females. The lines on the top of the bars show one standard deviation. Asterisks indict P values. * $P \leq 0.05$, **$P \leq 0.01$, *** $P \leq 0.001$, and ns non-significant. The result is significant across

the three datasets.
(EPS)

**S7 Fig. The increase in happiness, conditional on the co-occurrence with the other gender, is higher for female than for male characters. This Finding is Robust across Time Periods.** We analyze three datasets, including 6,087 movie synopses (**a**), 1,109 movie scripts (**b**), and 7,226 books (**c**). The increase in happiness conditional on the co-occurrence with the other gender across different times, measured in the average of positive regression coefficients k, are shown as bars (blue for males and orange for females). The lines on the top of the bars show one standard deviation. Asterisks indict P values. $^{*}$ P $\leq$ 0.05, $^{**}$P $\leq$ 0.01, $^{***}$ P $\leq$ 0.001, and ns non-significant. The result is significant across the three datasets.
(EPS)

**S8 Fig. The increase in happiness, conditional on the co-occurrence with the other gender, is higher for female than for male characters. This Finding is Robust across Genres.** We analyze three datasets, including 6,087 movie synopses (a), 1,109 movie scripts (b), and 7,226 books (c). The increase in happiness conditional on the co-occurrence with the other gender for various types of movies and books, measured in the average of positive regression coefficients k, are shown as bars (blue for males and orange for females). In the Gutenberg book dataset, T represents technology, A represents general work, F represents Local History of the Americas, D represents World History and History of Europe, Asia, Africa, Australia, New Zealand, etc., C represents Auxiliary Sciences of History, Q represents Science, B represents Philosophy, Psychology, Religion, P represents Language and Literatures. The lines on the top of the bars show one standard deviation. Asterisks indict P values. $^{*}$ P $\leq$ 0.05, $^{**}$P $\leq$ 0.01, $^{***}$ P $\leq$ 0.001, and ns non-significant. The result is significant across the three datasets.
(EPS)

**S1 File. Explanation of S1–S8 Figs.** We analyze overall emotion, negative emotion, a life unpacked during the co-occurrence, controlling story genres and periods and robustness check of regression models.
(DOCX)

# Acknowledgments

The authors thank Da Xiao (Beijing University of Posts and Telecommunications and Color Cloud Technology), Shiyu Zhang (The University of Michigan), Mingxia Chen (Tencent Research Institute), Qiu Yu (Minzu University of China) and Guangxue Wen (University of Pittsburgh) for inspiring discussions.

# Author Contributions

**Conceptualization:** Lingfei Wu, Cheng-Jun Wang.

**Data curation:** Huimin Xu, Zhang Zhang, Cheng-Jun Wang.

**Funding acquisition:** Cheng-Jun Wang.

**Investigation:** Huimin Xu, Zhang Zhang, Lingfei Wu.

**Methodology:** Huimin Xu, Lingfei Wu.

**Resources:** Cheng-Jun Wang.

**Visualization:** Huimin Xu, Zhang Zhang, Lingfei Wu.

**Writing – original draft:** Huimin Xu, Zhang Zhang, Lingfei Wu, Cheng-Jun Wang.

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
