## [Decision Letter · Decision Letter 0]

22 Jul 2019

PONE-D-19-16923

The Cinderella Complex: Word Embeddings Quantify Gender Stereotypes in Movies and Books

PLOS ONE

Dear Dr Wang,

Thank you for submitting your manuscript to PLOS ONE. After careful consideration, we feel that it has merit but does not fully meet PLOS ONE’s publication criteria as it currently stands. Therefore, we invite you to submit a revised version of the manuscript that addresses the points raised during the review process.

We would appreciate receiving your revised manuscript by Sep 05 2019 11:59PM. To enhance the reproducibility of your results, we recommend that if applicable you deposit your laboratory protocols in protocols.io, where a protocol can be assigned its own identifier (DOI) such that it can be cited independently in the future. For instructions see: http://journals.plos.org/plosone/s/submission-guidelines#loc-laboratory-protocols

We look forward to receiving your revised manuscript.

Kind regards,

Ilya Safro, Ph.D.

Academic Editor

PLOS ONE

Journal Requirements:

2. Please ensure that you refer to Figures 4 and 8 in your text as, if accepted, production will need this reference to link the reader to the figure.

Additional Editor Comments:

Dear Authors,

Thank you very much for submitting your paper for publication with PLOS One. Both reviewers agree that the paper requires a major revision. Please read carefully and address all their comments.

Best regards,

Ilya Safro

Reviewers' comments:

Reviewer's Responses to Questions

**Comments to the Author**

1. Is the manuscript technically sound, and do the data support the conclusions?

Reviewer #1: Partly

Reviewer #2: Yes

2. Has the statistical analysis been performed appropriately and rigorously? 

Reviewer #1: No

Reviewer #2: Yes

3. Have the authors made all data underlying the findings in their manuscript fully available?

Reviewer #1: Yes

Reviewer #2: Yes

4. Is the manuscript presented in an intelligible fashion and written in standard English?

Reviewer #1: Yes

Reviewer #2: Yes

5. Review Comments to the Author

Reviewer #1: The paper presents a computational analysis of gender stereotypes in movie synopses, movie scripts, and books from the project Gutenberg. Three types of analysis are performed:

1. Comparison of happiness changes for male and female characters when they interact;

2. The influence of happiness changes for male and female characters on movie quality and popularity;

3. Comparison of word usage in movies with the leading male or female character.

The paper studies an important problem of gender stereotypes that persist over the years in English fiction books and movies. It presents several interesting analyses that shed light on the problem from different perspectives. However, the authors tend to over-generalize their conclusions. For example, the higher increase in happiness for female characters may be due not to the presence of a male character (“the Prince”), but to the overall tendency of describing female characters with more emotional words. The authors can compare the emotional changes (increases and decreases in happiness) for female and male characters over the whole course of the story (not only when the two characters interact). Similarly in Sec. 3 (Unpacking the lives of female and male characters), the authors can analyze all the words describing the male and female characters to discover the main clusters/topics/communities.

Further, many details of the analyses need clarification:

- How do you measure happiness for different characters (male and female) in the same sentence/paragraph where both characters appear (co—occurrence)?

- Fig. 2-4: Do you take into account only positive slopes (as mentioned in P. 4)? Do both characters have to have a positive slope in emotional change for the text snippet to contribute to the average? Why don’t you consider negative slopes? Women can be in general described with more emotional words.

- It would be beneficial to provide statistical significance for all comparisons in the paper (e.g., in Fig. 2, 3, 4, 7).

- Fig. 2: “The increase in happiness is quantified by the weighted average of positive OLS regression coefficients” -> what are the weights? How do you get them?

- Table 1: please provide more explanations. Do you fit one regression model with all these variables? What do these variables mean? What do the stars mean? The gender of the leading character has a big impact on both quality and popularity of the movie. It will be interesting to see the impact of emotional change separately on movies with a male or female leading character.

- Sec. 3, p. 6: “… we separate the movie synopsis data into female and male groups” -> How? Based on the gender of the main character?

- Sec. 3, p. 6: why do you look for words “before and after the co-occurrence of characters to construct word co-occurrence network”? How do you separate which words describe which character? Why don’t you simply look at words around a single character (male or female)?

The authors need to at least acknowledge that there are many sources of potential errors in their analysis pipeline, e.g.:

- Automatic process of obtaining happiness scores is error-prone (correlation with manually obtained scores is 0.53);

- Happiness scores of individual words can change dramatically in some contexts (e.g., in the presence of negation);

- Automatic process of main character and their gender identification is error-prone.

It would be interesting to analyze gender stereotypes over time, especially in movies, to see if there have been any changes/improvements in the last years.

Minor comments:

- The paper needs thorough proof-reading.

- The figures are of low quality and hard to read.

- P. 4: “Figure 7” -> “Figure 3”?

- Fig. 1: name highlighting is wrong.

- Fig. 5 and 6: legends are almost the same, need fixing. Fig. 6: “female” -> “male”? Also, 1,750 (female) + 4,337 (male) = 6,087 movie synopsis. What about the rest of 6,657 movie synopsis?

Reviewer #2: The manuscript "The Cinderella Complex: Word Embeddings Quantify Gender Stereotypes in Movies and Books" examined whether and how much the so-called Cinderella complex is present in books and movie synopsis and scripts. Specifically, the authors aimed to identify how stories of movies and book contribute to perpetrate gender stereotypes. I think that this paper has many merits, in specific the authors conducted rigorous analyses of the texts that allowed to identify how stories contain a specific representation of women. However, I think that the manuscript needs to be deeply revised to improve its quality. In the following I will elaborate my major concerns.

Introduction

I have two major points that should be addressed in the introduction.

The first is related to gender stereotypes. They are simply cited with no explanation of what they actually are. Which are the contents of gender stereotypes and which are they functions? Social Psychological research have clearly shown that there are stereotypical expectations according to which women should be warm, communal, sociable, kind, etc, whereas women should be competent, agentic, skilled and assertive (for reviews see, Ellemers, 2018; Eagly, Wood, & Diekman, 2000). These stereotypes derive from the social roles historically attributed to men and women and contribute to maintain gender inequality in the society (especially in the workplace) at different levels. I think that this should be considered I the introduction and also in the general discussion to improve the broad quality of the paper. Indeed, gender stereotypes are very often mentioned by never explained deeply. Also, I wonder whether the Cinderella effect can be considered a stereotype. If yes (and I am not completely sure about it) the authors should motivate this possibility considering relevant scientific literature. In this vein, I would ask to motivate with more reference if the Cinderella complex could be considered a theory. Is this really a theory? For which scientific area? In the GD it should be also discussed how the Cinderella complex in movies and book can contribute to perpetrate gender stereotypes and inequality.

Second, when presenting the study, the authors anticipate the results instead of clarifying the hypotheses of their study: “By analysing a large size of stories using word embeddings, we discovered that the so-called Cinderella complex widely exists in movies and novels. In the following sections, we first present our analysis on the case of Cinderella”. They should explicitly state what they expect from the specific analyses and why (the reasons behind these hypotheses clearly grounded on the scientific literature).

Method

The authors should provide more information on how the material has been selected. They wrote “rigorously” but with which criteria? I do not understand how they arrived at the number of 16255 movies. And from these only 6657 were selected with the criteria of the length of the synopsis? The same comment applies to books.

As far as the “fortune” measure, did they use only the words cited in the text (i.e., success, happy, lucky, and failure, sad, unlucky)? If this is the case, this should be extensively motivated, because many other words could be related to the “fortune” construct. Importantly, why “fortune”? This should be anticipated in the theoretical part of the paper. How the aim of the study to look for the Cinderella complex in story telling is achieved by analysing the “fortune” of the character? This is a major point that the authors should address.

Why does the first study compare Cinderella and Forrest Gump? How Forrest Gump was chosen? It seems to me that they are not comparable for many reasons.

Results

The regression table is not clear to me. Beta, F and p values are missing. What are the values in parentheses? Moreover, how many models were run? Moreover, I do not fully understand why the result showing a higher relation between happiness of women and popularity could be interpreted as in line with the Cinderella complex or at least the fact that gender stereotypes are appreciated. This should be clearer commented.

It could be interesting also examining the relation between the number of co-occurrences and increase happiness.

Discussion

The specific and general implications of the results are poorly discussed. For instance, why books and movies with male characters include more verbs than those with female main characters? Here I would like to see a discussion concerning, for example, the agency characteristics stereotypically attributed to men and this result. Moreover, why adjectives are more represented for female characters? How this result could be interpreted? One possibility is that women are described in more abstract terms related to their traits and therefore are represented in a more complex way than men? Moreover, adjectives lead to inferences of stability of these traits, which are perceived as difficult to change (See Maass, 1999). In this line of reasoning, which are the implications for gender stereotypes?

At the beginning of the manuscript, the authors write about moral tales. Could this argument be discussed in the GD in order to explain how morality is related to gender stereotypes and the role of movies and book to perpetrate gender inequality though these morality norms?

Finally, it should be interesting some comments related to language use and the content of gender stereotypes, that is how language contribute to the maintenance of these stereotypes (see Menegatti and Rubini, 2017).

Minor points:

The numbers of pages are missing, they should have been added to facilitate the reading and revision. Manuscript text should also be double-spaced.

The manuscript ends with “etc”. This should be avoided.

References

Eagly, A. H., Wood, W., & Diekman, A. B. (2000). Social role theory of sex differences and similarities: A current appraisal. In T. Eckes & H. M. Traunter (Eds.), The developmental social psychology of gender (pp. 123–174). Mahwah, NJ: Lawrence Erlbaum.

Ellemers, N. (2018). Gender stereotypes. Annu. Rev. Psychol. 69, 275–298.

Maass, A. (1999). Linguistic intergroup bias: Stereotype perpetuation through language. In M. P. Zanna (ed.), Advances in Experimental Social Psychology (Vol. 31, pp. 79121). San Diego, CA: Academic Press.

Menegatti, M. & Rubini, M. (2017). Gender bias and sexism in language. In Oxford Research Encyclopedia of Communication. Oxford University Press.

6. PLOS authors have the option to publish the peer review history of their article (what does this mean?). If published, this will include your full peer review and any attached files.

Reviewer #1: No

Reviewer #2: No

---

## [Author Response · Author response to Decision Letter 0]

13 Sep 2019

Please kindly refer to the attached file titled Response to Reviewers.

---

## [Decision Letter · Decision Letter 1]

17 Oct 2019

PONE-D-19-16923R1

The Cinderella complex: Word embeddings reveal gender stereotypes in movies and books

PLOS ONE

Dear Dr Wang,

Thank you for submitting your manuscript to PLOS ONE. After careful consideration, we feel that it has merit but does not fully meet PLOS ONE’s publication criteria as it currently stands. Therefore, we invite you to submit a revised version of the manuscript that addresses the points raised during the review process.

We would appreciate receiving your revised manuscript by Dec 01 2019 11:59PM. To enhance the reproducibility of your results, we recommend that if applicable you deposit your laboratory protocols in protocols.io, where a protocol can be assigned its own identifier (DOI) such that it can be cited independently in the future. For instructions see: http://journals.plos.org/plosone/s/submission-guidelines#loc-laboratory-protocols

We look forward to receiving your revised manuscript.

Kind regards,

Ilya Safro, Ph.D.

Academic Editor

PLOS ONE

Reviewers' comments:

Reviewer's Responses to Questions

**Comments to the Author**

1. If the authors have adequately addressed your comments raised in a previous round of review and you feel that this manuscript is now acceptable for publication, you may indicate that here to bypass the “Comments to the Author” section, enter your conflict of interest statement in the “Confidential to Editor” section, and submit your "Accept" recommendation.

Reviewer #1: (No Response)

Reviewer #2: All comments have been addressed

2. Is the manuscript technically sound, and do the data support the conclusions?

Reviewer #1: Yes

Reviewer #2: Yes

3. Has the statistical analysis been performed appropriately and rigorously? 

Reviewer #1: Yes

Reviewer #2: Yes

4. Have the authors made all data underlying the findings in their manuscript fully available?

Reviewer #1: Yes

Reviewer #2: Yes

5. Is the manuscript presented in an intelligible fashion and written in standard English?

Reviewer #1: Yes

Reviewer #2: Yes

6. Review Comments to the Author

Reviewer #1: I would like to thank the authors of the manuscript for their substantial efforts at addressing all the reviewers’ concerns. I found the additional experiments presented in Figures S1-S6 very helpful. I believe their inclusion into the manuscript will strengthen the paper. Unfortunately, right now they are left only as figures in the Supplement with no explanation and no reference in the main body. It would be great to include these experiments and describe what was done, why, and what conclusions can be drawn from them in the main body of the paper or at least in the Supplemental Materials. Also, the authors provided to the reviewers detailed explanations for the experiments, but not all of these explanations found their way into the main manuscript. In particular, I think it is important to mention that happiness score normalization is done within a character, and what experiments were performed to confirm robustness of the results in Table 1 to story intensity and the gender of the leading character.

There are still some missing details:

- “We accumulate the happiness curve across sentences or paragraphs …” (p. 20) and “We then sum the averaged happiness scores across sentences … to obtain the emotion curves” (p. 6). What do you mean by “accumulate the curve”? Do you really sum the scores across sentences or do you just plot the scores for all the sentences? Do you do any sort of curve smoothing?

- Fig. 8a: Why are the frequency numbers (y axis) less than 1 (negative powers of 10) for movie synopsis?

Phrasing:

- “… men are more likely to use verbs than women” (p. 5) and “Males use more verbs than females” (Fig. 6). I think you mean that men are more likely than women to be described using verbs.

Reviewer #2: (No Response)

7. PLOS authors have the option to publish the peer review history of their article (what does this mean?). If published, this will include your full peer review and any attached files.

Reviewer #1: No

Reviewer #2: Yes: Michela Menegatti

---

## [Author Response · Author response to Decision Letter 1]

28 Oct 2019

We are grateful for the opportunity to revise the manuscript in response to these thoughtful reviews, and we believe that the paper is much stronger for having incorporated the suggestions from the reviewers.

---

## [Editor Report · Decision Letter 2]

5 Nov 2019

The Cinderella complex: Word embeddings reveal gender stereotypes in movies and books

PONE-D-19-16923R2

Dear Dr. Wang,

We are pleased to inform you that your manuscript has been judged scientifically suitable for publication and will be formally accepted for publication once it complies with all outstanding technical requirements.

With kind regards,

Ilya Safro, Ph.D.

Academic Editor

PLOS ONE
---

## [Editor Report · Acceptance letter]

15 Nov 2019

PONE-D-19-16923R2 

The Cinderella complex: Word embeddings reveal gender stereotypes in movies and books 

Dear Dr. Wang:

I am pleased to inform you that your manuscript has been deemed suitable for publication in PLOS ONE. Congratulations! Your manuscript is now with our production department. 

With kind regards,

on behalf of

Dr. Ilya Safro 

Academic Editor

PLOS ONE